# Pre-Germination Treatments at Operational Scale for Six Tree Species from the Sclerophyll Forest of Central Chile

**DOI:** 10.3390/plants11050608

**Published:** 2022-02-24

**Authors:** Eduardo Cartes-Rodríguez, Carolina Álvarez-Maldini, Manuel Acevedo, Marta González-Ortega, Alejandro Urbina-Parra, Pedro León-Lobos

**Affiliations:** 1Centro Tecnológico de la Planta Forestal, Instituto Forestal–Sede Biobío, Camino a Coronel Km 7.5, San Pedro de la Paz 7770223, Chile; ecartes@infor.cl (E.C.-R.); macevedo@infor.cl (M.A.); mgonzale@infor.cl (M.G.-O.); aurbina@infor.cl (A.U.-P.); 2Instituto Ciencias Agroalimentarias, Animales y Ambientales (ICA3), Universidad de O’Higgins, Campus Colchagua, Ruta 90 Km 3, San Fernando 2840440, Chile; 3Grupo de Especialidad Recursos Genéticos, Centro de Investigación Regional La Platina, Instituto de Investigaciones Agropecuarias, Avenida Santa Rosa 11610, La Pintana, Santiago 8831314, Chile; pleon@inia.cl; 4Centro de Estudios Avanzados en Zonas Áridas (CEAZA), La Serena 1720256, Chile

**Keywords:** Mediterranean, nursery production, seeds, water imbibition

## Abstract

Sclerophyll forest in Mediterranean central Chile has been subjected to severe degradation due to anthropic disturbances and climate change and is in need of restoration. Since direct seeding is usually unsuccessful, we need to research seed propagation to produce plants for restoration. Our objective was to assess pre-germination treatments for six native woody species (*Acacia caven*, *Lithraea caustica*, *Quillaja Saponaria*, *Porlieria chilensis*, *Kageneckia angustifolia*, and *Ceratonia chilensis*) of the sclerophyll forest, considering its operational applicability and consequences for nursery plant production. Treatments were selected according to previous studies, and operational applicability in nurseries. Germination and level of seeds water imbibition were assessed. Results indicate that time for seed water imbibition is critical for germination in *A. caven*, *P. chilensis* and *K. angustifolia*, with an average germination of 90.2 ± 2.0%, 85.0 ± 4.7%, and 47.4 ± 2.3%, respectively. Gibberellin did not improve germination compared to water soaking in *Q. Saponaria*, *K. angustifolia* and *P. chilensis*. In addition, physical scarification is a suitable treatment for *L. caustica* and *C. chilensis*, instead of chemical scarification, avoiding handling toxic and corrosive compounds in nurseries. We recommend assessing seed water imbibition rates as a key factor for proper germination processes.

## 1. Introduction

There are five Mediterranean-type climate regions in the world that represent less than 5% of global surface but and are catalogued as biodiversity hotspots because they harbor 20% of endemic vascular plants of the planet [1,2]. These regions are characterized by higher levels of endemism of the vascular flora, which are adapted to seasonal water deficit and high summer temperatures [3]. One of these regions is found in central Chile, between 30° S and 36° S approximately; this represents a transition between the Atacama Desert and mixed temperate forest of south Chile, with shrub formations and sclerophyll forests that dominate the landscape [1,4,5].

The Mediterranean-type climate region of Chile has been subjected to strong anthropic pressures [6] due to changes in soil use, which has reduced the coverage, structure, and composition of this ecosystem, and this has been worsened by the effect of forest and soil degradation and bushfires [7,8,9,10]. These events have increased in severity during the last decade, as Chile has been facing a mega-drought since 2010 [11,12,13]. In recognition of the central role of forests for carbon storage in vegetation and soil [14,15,16], the United Nations (UN) has declared the 2021–2030 the “Decade of Ecosystem Restoration” as an important tool to mitigate the increasing loss of biodiversity and rise in CO_2_ emissions. Altamirano et al. [17] reveal the urgent need to restore forest ecosystems at a global scale, which represents the best option for the achievement of goals described by the UN. This is even more relevant for sclerophyll forests because it requires greater restoration efforts considering its actual degradation state [18].

Chile and 114 other countries have subscribed to a series of restoration commitments [19]. In the context of the Paris Agreement and the update of the Determined Contribution at National Level in 2020, Chile is committed to afforest 70,000 hectares of native species for the formation of a permanent forest cover and to restoration processes of 1,000,000 hectares of landscape [20]. However, Bannister et al. [21] identified three bottlenecks for successful forest restoration in Chile: (1) a lack of national plan for forest landscape restoration; (2), poor quality and low supply of native plants species, which was thoroughly described by Acevedo et al. [22]; and (3) poor results in the establishment phase. Likewise, León-Lobos et al. [23] describe a fourth bottleneck focused on the low availability and seed quality of native species, which impairs plant production and the achievement of restoration goals. In addition, Mediterranean species from central Chile present several issues that negatively affect seed supply such as low density and species diversity in seed banks; besides low germination after sampling from soil [24,25,26], this has proven to be even more severe in landscapes dominated by *Quillaja Saponaria* Molina (Quillajaceae) and *Lithraea caustica* (Mol.) Hook. and Arn. (Anacardiaceae) [25]. Besides supply problems, seeds from the sclerophyll forest are exposed to inadequate environmental conditions during seed germination and establishment. Summer season in Mediterranean climate imposes important levels of stress, due to higher temperatures and the absence of relevant precipitations [27,28]. In the context of central Chile there is little experimental evidence that native species produce seed without dormancy at the moment of seed dispersal [26]. Evidence indicates that physical dormancy imposed by a hard and impermeable seedcoat is present in native species such as *Acacia caven* (Molina) Molina (Fabaceae) and *L. caustica* [29], which can be prompted to germinate with mechanic scarification or acid application. Likewise, the development of fleshy fruits in species such as *L. caustica* and *Porlieria chilensis* can induce dormancy due to the presence of chemical inhibitors [30,31], and in natural environments germination can be triggered by the passage through the intestinal tract of frugivore species such as native fox *Pseudalopex culpaeus* (Mol.) [32]. In these cases, the removal of the pericarp, acids or gibberellin treatment can be applied at operational levels.

Despite that seed anatomy and the environmental conditions can shed light regarding dormancy in seeds of species from the sclerophyll forest, there is a lack of information regarding specific dormancy and the identification of suitable pre-germination treatments that maximize its germination [23,33,34]. Although higher germination is a pre-requisite to unlock the following bottlenecks for restoration [21], such treatments should consider their applicability by local nurseries, that in most cases lack the proper training to, for example, handle toxic chemicals for acid stratification such as sulfuric acid [22,35].

The objective of this research is to assess pre-germination treatments for six native woody species (*A. caven*, *L. caustica*, *Q. Saponaria*, *P. chilensis* I.M. Johnst., *Kageneckia angustifolia* D. Don (Rosaceae), and *Ceratonia chilensis* (Molina) Stuntz emend. Burkart (Fabaceae) of the sclerophyll forest from central Chile, considering their operational applicability and consequences for nursery plant production.

## 2. Results

### 2.1. Seed Characterization

From the estimation of the number of seeds per kilogram, it is observed that bigger seeds belong to *A. caven* with 7,554 ± 216 seeds kg^−1^, followed by *L. caustica* and *C. chilensis* with 23,066 ± 660 seeds kg^−1^ and 24,359 ± 543 seeds kg^−1^, respectively, and finally *P. chilensis* with 37,169 ± 2379 seeds kg^−1^ and *Q. saponaria* with 112,413 ± 1,396 seeds kg^−1^; seeds with the lower weight were observed in *K. angustifolia* with 136,935 ± 3855 seeds kg^−1^.

A significant effect was observed in the time of water imbibition on the moisture content of seeds for all species (all *p* < 0.0012). The seeds initial moisture content was on average 14.9 ± 7.6%, which corresponded to seed water content immediately after storage and before soaking. After soaking in water for 24 h, significant increase was observed in this variable in all evaluated species (all *p* < 0.0058), while only for *A. caven*, *P. chilensis* and *K. angustifolia* a significant increase in moisture content was observed after 48 h of soaking (all *p* < 0.0152) (Table 1).

### 2.2. Nurseries Survey

Surveyed nurseries represent the 41.3% (*n* = 6) of nation-wide plant production for the selected species in this study during the 2017 to 2019 seasons, where chemical and scarification treatments present higher restrictions at an operational scale. Restrictions are mainly related to technical capabilities, due to lack of knowledge about how to apply the treatments at an operational scale, and to a lesser degree due to restrictions in infrastructure and equipment (Table 2).

Among the most used pre-germination treatments in nurseries are soaking in water at room temperature and chemical scarification with sulfuric acid (Table 3) for the selected species. Soaking in water ranged between two and 72 h for most nurseries and species, while exposure to sulfuric acid varies between 30 and 180 min depending on the protocol of each nursery and species. Surveyed nurseries do not apply plant hormones as pre-germination treatments, stratification, or mechanical scarification, for the assessed species.

### 2.3. Germination Experiment

For all evaluated species, a significant effect was observed in the interaction between pre-germination and measurement time (MED × TRAT) (all *p* < 0.0430).

In *Q. saponaria* average germination at the end of the experiment reached 90.2 ± 2.0%; 30 days after sowing, no significant differences between pre-germination treatments were observed. After day 16 no significant increments in germination were observed (all *p* > 0.1745), with an average of 86.1 ± 1.2% without significant differences between pre-germination treatments (Figure 1a).

In *L. caustica*, at the end of the germination experiment (63 days after sowing), average germination reached at 45.4 ± 5.2%, and no significant differences between pre-germination treatments were observed (all *p* = 1). Chemical scarification with sulfuric acid induced a reduction in germination times of 33 and 23 days in comparison with control treatment and physical scarification, respectively. With chemical scarification, germination did not show significant increases after day 18 of sowing (38.1 ± 2.4%) (all *p* > 0.2327), while in control and physical scarification treatments, germination remained constant at day 51 (40.6 ± 6.3%) and 46 (38.4 ± 4.3%), respectively (Figure 1b).

In *A. caven* pre-germination treatments showed significant differences in germination (all *p* < 0.0001). At the end of the evaluation period (day 53 since sowing), seed soaking in water for 48 h germination was 70.1 ± 6.8%, which was significantly higher than with physical and chemical scarification that reached an average germination of 24.3 ± 6.5%, which did not show significant differences (*p* = 1) (Figure 1c). At day 25 after sowing, we did not observe significant increments in germination in the control treatment that reached an average of 61.1 ± 4.2% (all *p* > 0.9071).

Similarly, final germination of *P chilensis* seeds showed significant differences between treatments (all *p* < 0.0001). Physical scarification caused low germination (0.3 ± 0.2%) compared to control and the application of gibberellic acid. Thus, at the end of the germination period (63 days since sowing), no significant differences were observed between control treatment and gibberellic acid application at 200 mg L^−1^ (*p* = 1), with an average germination of 47.4 ± 2.3%. However, soaking in gibberellic acid decreased the germination period by 11 days compared to control treatment (Figure 1d).

In *K. angustifolia* no significant differences in germination were observed at the end of the experiment (30 days after sowing) between pre-germination treatments (all *p* > 0.8581), an average germination of 85.0 ± 4.7% was reached. At day 16 after sowing germination had not increased significatively in any treatments (*p* > 0.3008), with an average germination of 80.5 ± 6.0% (Figure 1e).

At the end of the evaluation period in *C. chilensis*, 53 days after sowing, pre-germination treatments showed a significant effect in germination (all *p* < 0.0001). Chemical and physical scarification induced a significantly higher germination (83.7 ± 11.0%) compared to the control (16.1 ± 3.0%). No significant differences were observed between chemical and physical scarification (*p* = 0.5807) (Figure 1f). In the case of chemical scarification, 11 days after sowing germination did not increase significantly (all *p* > 0.690), reaching 70.4 ± 9.4%. For physical scarification a similar pattern was observed at day 30 after sowing (all *p* > 0.3874), and 85.6 ± 2.6% of germination was observed. In the control a stable germination level of 11.1 ± 2.4% was observed at day nine after sowing (all *p* > 0.8457).

## 3. Discussion

### 3.1. Seed Characterization

A low water content after 48 h of imbibition was observed in *L. caustica*, *P. chilensis*, and *C. chilensis*, in relation to the other assessed species (Table 1), which can be attributed to physical dormancy, previously reported in *L. caustica* and *C. chilensis* [25]. In *P. chilensis*, despite that our results show higher water imbibition than the results reported by Cabello et al. [36], there is agreement that water imbibition reaches a peak after 48 h of soaking in water, indicating that seedcoat impermeability is not an impairment for germination in this species. Similar results regarding water imbibition were observed in *A. caven*, with increased water content after 48 h of soaking in water, which does not agree with results from previous research [25,37,38]. Specifically, Funes and Venier [38] indicate that low water imbibition and low germination in non-scarified seeds (0%) vs. scarified seeds (96.6 ± 1.6%) is evidence of physical dormancy imposed by an impermeable seed coat, which could act as protection against humidity and temperature fluctuations [39]. However, in the research of Funes and Venier [38], imbibition was evaluated only up to 24 h, while our results indicate that maximum imbibition was reached 48 h after soaking (Table 1). In addition, germination was assessed for five days whereas, according to our results, maximum germination is reached after 25 days (Figure 1c), indicating that while seed coat impermeability could be present it can be overcome with longer water imbibition and germination periods. Thus, longer imbibition times in *A. caven* could be linked to prevention of germination in sites with unpredictable or sporadic rainfall [40], a characteristic of its habitat in central Chile.

According to Hartmann and Kester [41], seed germination occurs with water content from 40% to 60%, which would indicate that the only seeds that did not reach the needed water content for germination are the ones from *C. chilensis*, and this could explain the low germination levels observed in the control treatment (Table 2, Figure 1f). In addition, desiccation tolerance of the species can also affect germination, which would be directly linked to the initial water content of seeds (Table 1). However, the seed storage behavior of species included in this study had not been experimentally evaluated. Most species from dryland environments, such as central Chile are likely to have desiccation tolerance seeds [42]. According to the Seed Information Database [43] seeds of these species should be desiccation tolerant (Orthodox).

### 3.2. Germination Experiment

High germination in *Q. saponaria* seed is in agreement with previous reports [44,45]. The absence of significant differences between pre-germination treatments (Figure 1a) indicate that *Q. saponaria* seeds do not present dormancy, which contradicts Figueroa and Jaksic [25], who indicate that the presence of an undetermined dormancy in *Q. saponaria* seeds and a physiological dormancy as proposed by Baskin and Baskin [46] according to Donoso and Cabello [47] results. However, no information regarding seed storage conditions was displayed by authors that mentioned an undetermined dormancy in this species, which could also affect germination. In addition, soaking in water for 24 h should be enough to achieve high water content in seeds (Table 1) and promote a high germination (90.2 ± 2.0%), which is a smaller imbibition time than the 72 h reported by Benedetti et al. [48]. According to nursery survey results, 83% of nurseries a proper treatment for seed germination.

In *L. caustica*, Donoso and Cabello [47] reported a germination capacity of 59% and recommended to treat seeds with sulfuric acid for at least three hours. Similar results were reported by [49] after remotion of seeds epicarp. Our results indicate that chemical scarification with sulfuric acid accelerates the germination process; however, it does not increase germination capacity in comparison to physical scarification or water soaking (Figure 1b), treatments that become in a suitable alternative to chemical scarification. However, these last alternatives extend the germination process to 46 and 51 days, respectively. These periods should be taken into consideration by nursery managers for sowing planification activities. According to our survey, at least 80% of nurseries apply a proper pre-germination treatment in this species.

In *A. caven* our results do not agree with previous reports [37,50,51], where seeds exposure to sulfuric acid before sowing promotes high germination (between 70% to 90%) as result of physical dormancy breaking of seeds [25]. However, germination observed after physical and chemical scarification treatments suggest deterioration of the seeds, questioning the existence of physical dormancy (Figure 1c). According to nursery surveys, we observed that at least 80% of nurseries applied chemical or physical scarification (Table 3) as a substitute for chemical scarification; these treatments could be simplified by water soaking for longer periods of time, such as 48 to 72 h.

According to Cabello et al. [36], *P. chilensis* seeds do not have physical dormancy related to impermeable seed coat, which agrees with our results regarding an increased water imbibition until 48 h since sowing. Despite that Loayza et al. [52] indicated that germination in response to physical scarification depends on seed provenance, our results showed that scarification with hot water caused a decrease in germination compared to control treatment (0.3 ± 0.2% vs. 47.4 ± 2.3%, respectively) (Figure 1d), suggesting that *P. chilensis* seed coat it is not impermeable to water and imbibition in hot water should have a detrimental effect on seed viability of this species (Figure 1d). Instead, Cabello et al. [36] reported that *P. chilensis* seeds present endogenous physiological dormancy, which could be broken with 60 days of cols stratification or soaking in 400 mg L^−1^ of gibberellic acid, reaching a germination capacity of 78.2%, higher than results obtained in this research. Although our gibberellic acid treatment was 200 mg L^−1^ and no differences were observed in germination compared to control, gibberellic acid decreased germination times by 11 days relative to control treatment, indicating that physiological dormancy could be present in this species. Despite that gibberellic acid application seems the most appropriate pre-treatment for this species, 100% of nurseries that produce *P. chilensis* only apply soaking in water as pre-germination and no specific treatment to break physiological dormancy is considered (Table 3).

Results observed in *K. angustifolia* regarding germination capacity agree with Takayashiki et al. [53], although the authors indicate a soaking in water for four days as pre-treatment, while according to our results 48 h in water is enough to achieve a water content of 105.4 ± 8.9% and to promote a germination of 85.0 ± 6.0%. These last results are consistent with several authors [54,55,56] and the surveyed nurseries that achieved germination between 70% to 80% without pre-germination treatments in direct sowing, indicating that *K. angustifolia* seeds are not dormant.

Scarification has been reported by several authors [57,58,59,60], as a method to break physical dormancy in *Prosopis* species seeds. In fact, it had been reported that passage through a digestive tract of frugivores and cattle induce germination by promoting seed coat rupture [61,62]. However, Vilela and Ravetta [60] indicated that chemical scarification for 15 min reduced germination *C. chilensis*, while physical scarification (dipped in boiling water until water reached room temperature) induced higher germination. In *Prosopis ferox*, a similar species, Ortega et al. [59] obtained higher germination after physical scarification (93.0 ± 0.03%) and chemical scarification (91.0 ± 0.02%) compare with hydrochloric acid (14.0 ± 0.02%). Our results agree with the statement that *C. chilensis* germination is promoted by chemical (with sulfuric acid) or physical scarification (Figure 1f). Physical and chemical scarification did not caused differences in germination, but affected the time when germination reached a stable value (30 vs. 9 days since sowing, respectively) (Figure 1f). Although 66% of surveyed nurseries apply a proper scarification treatment (physical or chemical, 33% each), the time needed to complete the germination process is a factor that should be considered at a large operational scale.

### 3.3. Operational Applicability of Pre-Germination Treatments

Among the main problems mentioned by nurseries, we highlight the lack of technical capabilities (Table 3); there is a knowledge breach regarding to preparation, application, and manipulation of some chemical products for the implementation of treatments. This lack of information could be amended through instances of training and technological transference instances. This agrees with the diagnosis reported by León-Lobos et al. [23] where lack of knowledge regarding the dormancy of breaking in seeds of several native species is identified as a bottleneck for the fulfilment of Chile restoration commitments.

In regard to chemical scarification, its operational implementation (Table 3) is limited in some nurseries by the manipulation of corrosive chemical such as sulfuric acid, and managers indicate concern regarding the risk to staff safety and chemical residue disposal [35]. On the other hand, physical scarification application is limited due to technical restrictions and lack of infrastructure (Table 3), in particular the need for equipment to process large numbers of seeds at operation scale, technology that is not widely distributed in Chilean nurseries.

## 4. Materials and Methods

### 4.1. Species Selection and Locations for Seed Collection

Six tree species from the sclerophyll forest were selected between Valparaíso and Biobío regions, three were selected according to dominance and three according to the degree of ecological vulnerability (for a full description of the species see Appendix A). Vulnerability was referred to the conservation state according to the Classification Regulation of the Species from the Environmental Ministry of Chile [63]. Regarding the dominance criteria *Q. saponaria*, *L. caustica* and *A. caven* were selected, while *P. chilensis*, (vulnerable), *K. angustifolia* (near threatened) and *C. chilensis* (vulnerable) were selected according to conservation criteria.

Seed collection was performed between January and March of 2020 in the populations indicated in Figure 2. Seeds were sampled from at least 10 trees for each species with a minimal distance of 15 m between each tree. Once collected, seeds were transferred to the Centro Tecnológico de la Planta Forestal from the Instituto Forestal (36°50.9′ S; 73°7.9′ W), Biobío region, Chile, for cleaning and storage at 4 °C until mid-May of 2020.

### 4.2. Seed Characterization

As part of seed characterization prior to pre-germination treatment application, the number of seeds per kilogram, the initial water content of seeds, and the increase in water seed imbibition was recorded, seed weight was recorded in three samples of 100 seeds (replicates) for each species to estimate the number of seeds per kilogram. Then, each sample was divided in two sub-samples of 50 seeds each, which were used to determine initial moisture content (*w*/*w*, dry basis), and after 24 and 48 h of soaking in distilled water. For this, one seed sub-sample of each species was weighted before soaking and after 24 and 48 h of soaking, while the other sub-sample was oven dried in a forced ventilation oven (Binder, model FD115, Tuttlingen, Germany) at 105 °C until constant weight. Weight of fresh and dried seeds were recorded on a 0.001 g precision scale (Quimis, Q DH-203, São Paulo, Brazil).

### 4.3. Pre-Germination Treatments

Three pre-germination treatments were evaluated for each species, which were selected according to available information from previous experiments [36,37,44,47,50,53,64,65,66,67,68,69,70] and information obtained from a survey performed to six nurseries to consider pre-germination treatments that were feasible to apply at operational scale at that were commonly applied in nurseries (Appendix A). Selected nurseries for the survey produced a larger amount of the native species selected for this research between 2017 and 2019 nationally [71].

The survey identified the pre-germination treatments operationally applied for each evaluated species, and restrictions for proper treatment application linked to technical capabilities, infrastructure, or equipment. In addition, knowledge gaps regarding the benefits of the application of pre-germination treatments was assessed.

For *L. caustica*, *A. caven*, and *C. chilensis* seeds with reported physical dormancy [25] two scarification treatments were applied: (1) chemical scarification, consisted of exposure of seeds to sulfuric acid (PQM Fermont, Monterrey, Mexico) at a concentration of 97.3% for 90 min, then seeds were rinsed with distilled water and soaked in water at room temperature for 48 h; (2) physical scarification, consisted of exposure of seeds to water at 80 °C, seeds were cooled until room temperature and soaked in water for 48 h. For *Q. Saponaria* and *K. angustifolia*, pre-germination treatments consisted of the use of gibberellin to break physiological dormancy, seeds were soaked in gibberellin at 200 mg L^−1^ and 600 mg L^−1^ (GA_3_, Giberplus, Anasac, Santiago, Chile) for 48 h. In the case of *P. chilensis*, treatments were the use of gibberellin at 200 mg L^−1^ and physical scarification as previously described. For all species, seed soaking in distilled water for 48 h corresponded to control treatment.

### 4.4. Germination Experiment

The germination experiment was performed in the Centro Tecnológico de la Planta Forestal of the Instituto Forestal, in greenhouse conditions (36 m^2^). Maximum air temperature was limited to 25 °C through a forced ventilation system. A photoperiod of 12 day and 12 dark was established with six halide lamps of 400 Watts each (Philips Master HPI-Plus, Brussels, Belgium). To characterize environmental conditions during germination, air temperature (°C) and relative humidity (%) was measured with an Atmos 14 sensor (METER Group Inc., Pullman, WA, USA) and for substrate temperature (°C) a RT-1 (METER Group Inc.) sensor was used. Environmental data were recorded every 30 min with a ZL6 datalogger (METER Group Inc.). During the germination experiment an average air temperature of 19.3 ± 4.4 °C was observed, with a daily thermal oscillation of 10.2 ± 4.5 °C. Minimum and maximum daily substrate temperatures were 15.1 ± 3.0 °C and 22.4 ± 2.7 °C, respectively. Air relative humidity ranged between 35% and 84%, with an average of 63 ± 9%.

Sowing for six species was performed in June of 2020 in expanded polystyrene trays of 0.13 L (15 cm depth) and 84 cavities (336 cavities m^−2^). Three seeds were sowed in each cavity at 1 cm depth approximately, three trays (replicates) were sowed for each germination treatment. Composted *Pinus radiata* bark was used as substrate with particles smaller than 10 mm (pH = 5.3 ± 0.01; organic matter = 76.1 ± 2.8%; total nitrogen = 0.9 ± 0.1%; C/N relation = 48.5 ± 6.4; N-NO_3_ = 233.6 ± 37.3 mg kg^−1^; N-NH_4_ = 662.6 ± 56.3 mg kg^−1^; water retention = 0.45 m^3^ m^−3^). Irrigation was performed with watering cans once a day, maintaining high humidity in the surface of the substrate.

Germination was measured three times per week (Monday, Wednesday and Friday), the number of germinated seeds was recorded out of the total sowed seeds (84 cavities × 3 seed cavities^−1^) for each of the three trays (replicates), species and pre-germination treatment. Since the sowing, germination was monitored for 30 days in *Q. Saponaria* and *K. angustifolia*, 53 days in *A. caven* and *C. chilensis*, and 63 days in *L. caustica* and *P. chilensis*. Occurrence of germination was considered when the epicotyl emerged from the substrate surface.

### 4.5. Data Analysis

The average values of number of seeds per kilogram was calculated for each species from the weight of 100 seeds. The analysis of relative water content of seeds was performed through a repeated measurement analysis of one way for each species, considering the time for seed water imbibition time as factor (MED). Environmental data collection allowed the estimation of average temperature, minimum and maximum daily temperatures in air and substrate, and average relative humidity during the germination period.

For the germination, the experimental design corresponded to a completely randomized design with three pre-germination treatments (TRAT) for each species and with three replicates for each treatment. For each species, a germination analysis was performed through a repeated measures analysis of two ways for measurement time (MED) and pre-germination treatment (TRAT), assessing the main effects and interactions.

Repeated measure analysis was performed through a generalized mixed model using PROC GLIMMIX (SAS Institute, Cary, NC, USA) with selection of distribution and structure of the variance-covariance residual considering the Akaike Information Criteria (normal distribution and unstructured variance and covariance matrix for every analysis). Multiple comparison tests were performed for significant effects according to Tukey.

The time during germination with no further significant differences in the proportion of germinated seeds was evaluated performing comparison tests considering significant effects in the variance model (MED × TRAT).

Graphs development for data visualization were designed with SigmaPlot 10.0 software (Systat Software Inc., Chicago, IL, USA).

## 5. Conclusions

Although several nurseries apply the proper pre-germination treatment in some species such as *Q. saponaria*, *P. chilensis* and *K. angustifolia*, there is room for improvement in applied treatments in the rest of the species. For example, in *A. caven* the water imbibition time should be considered to improve germination before the application of other pre-germination treatments. In addition, in *L. caustica* and *C. chilensis* chemical scarification could be replaced by physical scarification to avoid the issues linked to manipulation of chemicals.

The evaluation of the rate of water imbibition is needed for the evaluated species before the implementation of required pre-germination treatments at an operational scale to avoid seed germination decay.

Physical scarification with hot water (80 °C) is a proper alternative to chemical scarification. However, higher germination times should be considered in sowing calendarization at a large operational scale.

Extension and technical transference instances could be helpful to reduce the breach in knowledge indicated by national nurseries. This could help to achieve an optimal implementation of pre-germination treatments and to avoid the application of incorrect treatments such as in *A. caven*, *P. chilensis*, and *C. chilensis*, which could generate high losses in seed supply. These actions tackle the bottleneck related to the lack of a proper seed supply to achieve reforestation commitments in Chile.

## Figures and Tables

**Figure 1 plants-11-00608-f001:**
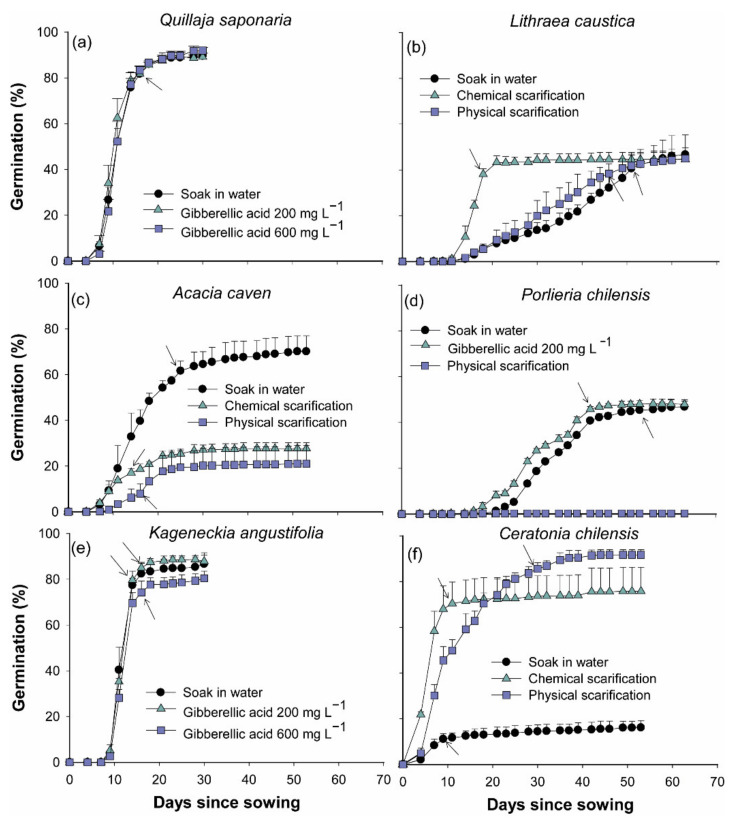
Germination (%) observed in *Quillaja Saponaria* (**a**), *Lithraea caustica* (**b**), *Acacia caven* (**c**), *Porlieria chilensis* (**d**), *Kageneckia angustifolia* (**e**), and *Ceratonia chilensis* (**f**) according to different pre-germination treatments. Arrows indicate the day after sowing where no further increment in germination was observed. Symbols indicate mean + s.d.

**Figure 2 plants-11-00608-f002:**
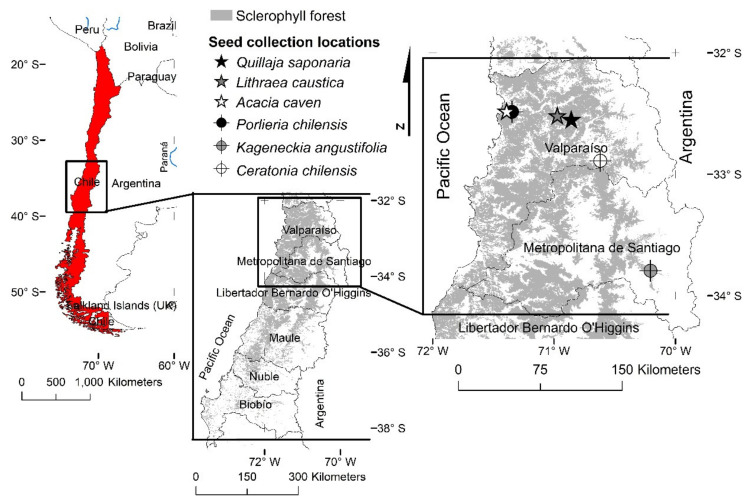
Distribution of sclerophyll forest (in grey) in Mediterranean central Chile and seed collection locations for *Acacia caven*, *Quillaja saponaria*, *Lithraea caustica*, *Porlieria chilensis*, *Kageneckia angustifolia* and *Ceratonia chilensis*.

**Table 1 plants-11-00608-t001:** Seed moisture content (*w*/*w* dry basis, g g^−1^) initial and after soaking in water for 24 and 48 h of sclerophyll species of central Chile (mean ± s.d.; *n* = 3). Letters indicate significant differences in moisture content for each species, between the different measurement times (Tukey, *p* < 0.05).

Species	Seed Moisture Content (g g^−1^)		
	Initial	Soaking in Water 24 h	Soaking in Water 48 h
*Quillaja saponaria*	0.139 ± 0.042 ^b^	0.851 ± 0.108 ^a^	0.935 ± 0.075 ^a^
*Lithraea caustica*	0.097 ± 0.023 ^b^	0.522 ± 0.040 ^a^	0.536 ± 0.021 ^a^
*Acacia caven*	0.273 ± 0.056 ^c^	0.627 ± 0.062 ^b^	0.903 ± 0.098 ^a^
*Porlieria chilensis*	0.182 ± 0.077 ^c^	0.458 ± 0.065 ^b^	0.626 ± 0.074 ^a^
*Kageneckia angustifolia*	0.101 ± 0.030 ^c^	0.867 ± 0.105 ^b^	1.054 ± 0.089 ^a^
*Ceratonia chilensis*	0.101 ± 0.027 ^b^	0.217 ± 0.055 ^a^	0.262 ± 0.046 ^a^

**Table 2 plants-11-00608-t002:** Restrictions identified in surveyed nurseries for application of pre-germination treatments at operation scale in evaluated species. (^1^): Gibberellic acid; (^2^) Hot water; (^3^) Sulfuric acid.

Pre-Germination Treatment	Without Restrictions (%)	Restrictions Due to Capabilities Techniques (%)	Restrictions Due to Infrastructure and/or Equipment (%)	Don’t Know the Benefits (%)
Water Soaking (RT)	100	-	-	-
Plant Hormone (^1^) *	67	16.5	16.5	-
Wet-cold Stratification	67	16.5	-	16.5
Physical Scarification (^2^)	100	-	-	-
Mechanical Scarification	50	33	17	-
Chemical Scarification (^3^)	50	33	17	-

* Gibberellic acid concentrations were not declared by the nurseries surveyed.

**Table 3 plants-11-00608-t003:** Pre-germination treatments applied by nurseries at operational scale, for the species evaluated in the sclerophyll forest. (^1^): Hot water; (^2^): Sulfuric acid.

Species	Soaking in Water (%)	Physical Scarification (%) (^1^)	Chemical Scarification (%) (^2^)	Soaking in Coke^®^ (%)	Direct Sowing (%)
*Quillaja saponaria*	83	0	0	0	17
*Lithraea caustica*	0	20	60	20	0
*Acacia caven*	0	20	60	20	0
*Porlieria chilensis*	100	0	0	0	0
*Kageneckia angustifolia*	100	0	0	0	0
*Ceratonia chilensis*	34	33	33	0	0

## Data Availability

The datasets generated for this study are available on request to corresponding author.

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
