# Peer review of "Pre-Germination Treatments at Operational Scale for Six Tree Species from the Sclerophyll Forest of Central Chile"

_plants, 2022, doi:10.3390/plants11050608_

Round 1
Reviewer 1 Report
Dear Editor, Dear Authors,
I am glad to be able to review this manuscript entitled "Pre-Germination Treatments at Operational Scale for Six Tree Species from the Sclerophyll Forest of Central Chile". The conducted experiment and its results are very interesting for me. The effective method of breaking the dormancy of seeds, especially trees and shrubs, is still one of the most popular issues in modern seed science. Therefore, I believe the Plants Editorial Board should consider publishing this manuscript.
Below are some notes that will help the Authors to improve this manuscript:
1) Please think about the shortening of the Abstract, I find it long-winded, especially the first part of it.
2) Keywords should be in alphabetical order.
3) There should not be enlarged spaces between paragraphs.
4) In my opinion, the pattern of the experiment is good and properly described. The only information missing here is on seed moisture and air humidity during seed storage (L316).
5) Sub-chapter titles - the first letters in words should be capital letters.
6) Tables and Figures titles - smaller font size - 9.
7) Table 1 - letters incicate significant differences do not bold; column headers - the first letters should be uppercase; the top and bottom lines should be thicker.
8) Tables 2 and 3 - column headings - the first letters should be uppercase; the top and bottom lines should be thicker; do not include a space between the value and %.
9) Figures 1 and 2 - they have very poor resolution and are illegible, especially Figure 2.
10) The entire manuscript must be carefully revisited and carefully adapted to the Plants template. Please pay particular attention to the References chapter, where journal abbreviations should be used; some of the publications are quite old, maybe remove some of them?
Author Response
Reviewer 1
- Please think about the shortening of the Abstract, I find it long-winded, especially the first part of it.
R: The abstract was shortened and adjusted to Plants requirements.
- Keywords should be in alphabetical order
R: Keywords were rearranged in alphabetical order.
- There should not be enlarged spaces between paragraphs
R: The manuscript was revised and enlarged spaces between paragraphs were removed.
- In my opinion, the pattern of the experiment is good and properly described. The only information missing here is on seed moisture and air humidity during seed storage (L316).
R: The air humidity and seed moisture were not measured during seed storage. However, the reported initial value of seed moisture on table 1, corresponds to seed water content immediately after storage and before water soaking.
- Sub-chapter titles - the first letters in words should be capital letters
R: Sub-chapter titles have been modified accordingly
- Tables and Figures titles - smaller font size – 9
R: Font size of figures and tables were change to 9.
- Table 1 - letters indicate significant differences do not bold; column headers - the first letters should be uppercase; the top and bottom lines should be thicker.
R: Requested changes were made in Table 1.
- Tables 2 and 3 - column headings - the first letters should be uppercase; the top and bottom lines should be thicker; do not include a space between the value and %
R: Requested changes were made in Table 2 and Table 3.
- Figures 1 and 2 - they have very poor resolution and are illegible, especially Figure 2.
R: Resolution of both figures was improved.
- The entire manuscript must be carefully revisited and carefully adapted to the Plants template. Please pay particular attention to the References chapter, where journal abbreviations should be used; some of the publications are quite old, maybe remove some of them?
R: The reference list was revised for errors and corrected. Some of the references are old because sadly those are the only available information regarding the species in this research.
Reviewer 2 Report
Authors investidated pre-germination treatments for six native woody species (Acacia caven, Lithraea caustica, Quillaja Saponaria, Porlieria chilensis, Kageneckia angustifolia, and Ceratonia chilensis of the sclerophyll forest from Chile in terms of succesful nursery plant production and the urgency to restore forest ecosystems at a global scale.
MAJOR POINTS:
1. Authors measured the initial water content of seeds, please show this data. Also add the information whether seeds are desiccation tolerant or desiccation sensitive. In this context the initial water content of seeds can affect their germination.
Proper water content during 3 months of storage is crucial for high germination. Some seeds must be dried before storage, other watered. Authors skipped this problem. Please indicate whether the water content was reduced to the optimal for each species?
2. Graphs/tables
Table 2:
transfer units to names of columns/rows
change "soakinh" to "soaking"
change "(room T°)"; use RT or give specific tempearture and then add "C"
Figure 1:
Consider changing graphs from black-and-white to color graphs and use one specific colort for each pre-treatminet, because for example the grey color is in the legend specified as chemical scarification or gibberelic acid. This figure is a panel of graps, therefore unified legend color would be useful for clear interpretation of the data.
Text in this figure is a little blurry, please check the resolution of this figure and increase it whether too low.
Consider the addition of a species anme at the top of each graph.
Figure 2:
text in this figure is also blurry, please increase the quality
3. Discussion Section:
In line 213 Authors wrote that "Q. saponaria seeds does not present dormancy" wheraas other literature data clearly state a dormancy. Authors used seeds which were stored at 4 °C for 3 months before analyses. This could affect the dormancy depth/presence. Include this infoprmation to the text.
Conclusion;
lines 293-301 look like a conclusion, which can be partially merged with this section at page 12
Minor points:
l.13 "Centro de Estudios Avanzados en Zonas Áridas (CEAZA), La Serena, Chile" is a hyperlink
l.31 "imbibition rated" or "imbibition rates"?
l.68 change the font type/size/format of "(Molina) Hook. & Arn."
l.90-93 bold format is neede here? The format of this paragraph is different fom these above.
l.93 transfer "2. Results" to separate line and format as heading
l. 120 "(2) Hot water; (3) 120 Sulfuric acid." whereas l.133 "(1): hot water; (2): sulfuric acid." - unify the format
l.265 "had been" or "it had been"
l.370 "ml" whereas concentraions of gibberelic acid is given per liter as 'L"
l.428-430 can be deleted
l.435 "314" seems to be unnecessary
Author Response
Reviewer 2.
- Authors measured the initial water content of seeds, please show this data. Also add the information whether seeds are desiccation tolerant or desiccation sensitive. In this context the initial water content of seeds can affect their germination. Proper water content during 3 months of storage is crucial for high germination. Some seeds must be dried before storage, other watered. Authors skipped this problem. Please indicate whether the water content was reduced to the optimal for each species?
R: Although there has not been a characterization of desiccation tolerance in these species, we added a sentence for clarification in line 210-216 was added.
Initial water content displayed in Table 1, corresponded to water content immediately after seed storage and before water soaking treatment, this was also clarified in line 101-102.
- Table 2:
transfer units to names of columns/rows
change "soakinh" to "soaking"
change "(room T°)"; use RT or give specific tempearture and then add "C"
R: Requested changes were done.
- Figure 1:
Consider changing graphs from black-and-white to color graphs and use one specific colort for each pre-treatminet, because for example the grey color is in the legend specified as chemical scarification or gibberelic acid. This figure is a panel of graps, therefore unified legend color would be useful for clear interpretation of the data.
Text in this figure is a little blurry, please check the resolution of this figure and increase it whether too low.
Consider the addition of a species anme at the top of each graph.
R: Figure 1 was changed as requested. Resolution was improved.
- Figure 2:
text in this figure is also blurry, please increase the quality
R: Quality of the image was improved.
- In line 213 Authors wrote that "Q. saponaria seeds does not present dormancy" wheraas other literature data clearly state a dormancy. Authors used seeds which were stored at 4 °C for 3 months before analyses. This could affect the dormancy depth/presence. Include this infoprmation to the text.
R: Information regarding storage conditions in this species was not delivered by previous authors that mentioned an unspecified dormancy in this species. This is now clarified in the discussion.
- lines 293-301 look like a conclusion, which can be partially merged with this section at page 12
R: Paragraphs were partially merged in the conclusions at page 12.
- 13 "Centro de Estudios Avanzados en Zonas Áridas (CEAZA), La Serena, Chile" is a hyperlink
R: Hyperlink was deleted.
- 31 "imbibition rated" or "imbibition rates"?
R: “rated” was changed to “rates”
- 68 change the font type/size/format of "(Molina) Hook. & Arn.
R: Font format was changed.
- 90-93 bold format is neede here? The format of this paragraph is different fom these above.
R: Bold font was deleted. The paragraph format was corrected.
- 93 transfer "2. Results" to separate line and format as heading
R: Format was corrected.
- 120 "(2) Hot water; (3) 120 Sulfuric acid." whereas l.133 "(1): hot water; (2): sulfuric acid." - unify the format
R: Format was unified
- 265 "had been" or "it had been"
R: “had been” was changed to “it had been”
- 370 "ml" whereas concentraions of gibberelic acid is given per liter as 'L"
R: 130 ml was changed to 0.13 L.
- 428-430 can be deleted
R: Sentence in lines 428 to 430 was deleted.
- 435 "314" seems to be unnecessary
R: 314 was deleted.
Round 2
Reviewer 2 Report
Authors improved the manuscript and responded to all my suggestions.